# α-L-Fucosidases from an Alpaca Faeces Metagenome: Characterisation of Hydrolytic and Transfucosylation Potential

**DOI:** 10.3390/ijms25020809

**Published:** 2024-01-09

**Authors:** Agnė Krupinskaitė, Rūta Stanislauskienė, Pijus Serapinas, Rasa Rutkienė, Renata Gasparavičiūtė, Rolandas Meškys, Jonita Stankevičiūtė

**Affiliations:** Institute of Biochemistry, Life Sciences Center, Vilnius University, Sauletekio av. 7, LT-10257 Vilnius, Lithuania; ruta.stanislauskiene@bchi.vu.lt (R.S.); pijus.serapinas@gmc.stud.vu.lt (P.S.); rasa.rutkiene@bchi.vu.lt (R.R.); renata.gasparaviciute@bchi.vu.lt (R.G.); rolandas.meskys@bchi.vu.lt (R.M.)

**Keywords:** α-L-fucosidases, transfucosylation, fucose-containing saccharides, fucosylated amino acids, human milk oligosaccharides

## Abstract

In various life forms, fucose-containing glycans play vital roles in immune recognition, developmental processes, plant immunity, and host-microbe interactions. Together with glucose, galactose, *N*-acetylglucosamine, and sialic acid, fucose is a significant component of human milk oligosaccharides (HMOs). Fucosylated HMOs benefit infants by acting as prebiotics, preventing pathogen attachment, and potentially protecting against infections, including HIV. Although the need for fucosylated derivatives is clear, their availability is limited. Therefore, synthesis methods for various fucosylated oligosaccharides are explored, employing enzymatic approaches and α-L-fucosidases. This work aimed to characterise α-L-fucosidases identified in an alpaca faeces metagenome. Based on bioinformatic analyses, they were confirmed as members of the GH29A subfamily. The recombinant α-L-fucosidases were expressed in *Escherichia coli* and showed hydrolytic activity towards *p*-nitrophenyl-α-L-fucopyranoside and 2′-fucosyllactose. Furthermore, the enzymes’ biochemical properties and kinetic characteristics were also determined. All four α-L-fucosidases could catalyse transfucosylation using a broad diversity of fucosyl acceptor substrates, including lactose, maltotriose, L-serine, and L-threonine. The results contribute insights into the potential use of α-L-fucosidases for synthesising fucosylated amino acids.

## 1. Introduction

Natural polysaccharides (or glycans) are unique because of the diversity of their structure and architecture, which gives them almost unlimited functionality. Some glycan monosaccharides are specific to taxonomic groups, but universal ones, such as fucose, are found in nearly all life forms [1,2,3]. Fucose is a hexose and an unusual monosaccharide because it has no hydroxy group at the sixth position. Moreover, in nature, fucose usually appears in the L-configuration. In contrast, the other hexoses found in glycans are predominantly in the D-configuration. In glycans, fucose is linked by a α1-2-, α1-3/4-, α1-6-, or *O*-glycosidic bond. Fucose-containing glycans are involved in many physiological processes, including immune recognition, development and neural functions, plant immunity, and host-microbe interactions. Fucose is found in microbial glycans, mucus glycans, plant, fungal, and algal polysaccharides. Also, it plays a significant role in the human body as a component of secretor blood group antigens and human milk oligosaccharides (HMOs) [4].

HMOs contain five monosaccharide molecules: glucose, galactose, *N*-acetylglucosamine, fucose, and sialic acid, usually *N*-acetylneuraminic acid. All HMOs contain lactose elongated by *N*-acetyllactosamine and lacto-*N*-biose units. The elongated structure can be modified linearly or branched, for example, fucosylated in α1-2, α1-3, or α1-4 manner. In addition, isomeric forms of various oligosaccharides, such as lacto-*N*-fucopentaose [5], are observed. Over 200 HMOs have been identified [5]. However, the composition of milk oligosaccharides produced by each woman can vary [6]. HMOs benefit infants by acting as prebiotics and supporting healthy bacteria growth. They also act as anti-adhesive antimicrobials, preventing the attachments of various pathogens and viral, bacterial, and protozoan infections. HMOs may even play a role in protecting breast-fed infants from HIV infection. This may explain why 80–90% of breastfed babies do not acquire the virus from their mothers [6,7]. The research continues to understand HMO’s role in the infection [8]. Moreover, these oligosaccharides interact directly with intestinal epithelial cells and modulate their gene expression [9,10]. They also modulate the immune response in neonates by affecting lymphocyte cytokine production [11]. Because of these benefits, HMOs are highly sought-after components of human milk formulas as alternatives to breast milk and as additives to functional foods [6,12].

No other mammal milk has such a wide structural diversity of milk oligosaccharides as human milk. Therefore, milk from other mammals, such as cows or goats, is a poor alternative for human infants. Currently, no HMOs are added to milk. Compounds such as galactooligosaccharides and fructooligosaccharides are added as HMO alternatives. However, fructose is not found as a human milk component, and neither fructooligosaccharides nor galactooligosaccharides are fucosylated; therefore, they cannot mimic the same benefits as HMOs [6]. Recently, 2′-fucosyllactose and lacto-*N*-neotetraose have been approved as food ingredients by official authorities [13]. However, with such a vast diversity of fucosylated oligosaccharides, researchers continue to work on developing efficient methods for synthesising these compounds.

HMO lactodifucotetraose was produced in *Escherichia coli* using fucosyltransferases with narrow acceptor selectivity [14]. 2′-Fucosylgalactose was synthesised using colanic acid and engineered *Saccharomyces cerevisiae* [15]. One-pot three-enzyme synthesis of lacto-*N*-fucopentaose was performed, with an excellent yield of 95%, using α-1,2-fucosyltransferase from *Thermosynechococcus* sp. NK55a [16]. Fucosyl-*N*-acetylglucosamine disaccharides of Fuc-α-1,3/1,6-GlcNAc were also produced with a yield of 79.0% by α-L-fucosidase from *Bacteroides fragilis* NCTC9343 [17]. Various other compounds, such as lacto-*N*-fucopentaose isomers, lacto-*N*-difucohexaose, difucosyllactose, lacto-*N*-neohexaose, fucosyllacto-*N*-neohexaose, and others were synthesised via application of α-L-fucosidases [18].

Several approaches are applied to produce fucosylated saccharides, ranging from chemical synthesis to chemoenzymatic or whole-cell catalysis. Chemical synthesis of these compounds is usually tiresome and requires protection and deprotection steps. This has a significant adverse impact on the yield of the final product. Moreover, regioselectivity is challenging to achieve [19]. An enzymatic approach can be employed to attain regioselective oligosaccharide and glycan synthesis. The enzymatic approach usually involves the application of fucosyltransferases or fucosidases. Employment of fucosyltransferases offers high regioselectivity and accurate glycoside synthesis. However, glycosyltransferases are generally not expressed in high yields and require complex and expensive sugar nucleotide substrates.

Moreover, multi-enzyme cascade systems are needed for nucleotide recycling to increase the cost efficiency of the process. Glycoside hydrolases, in this case, α-L-fucosidases, could provide an alternative to fucosylated oligosaccharide synthesis with the ability to operate with cheap and abundant substrates. α-L-Fucosidases catalyse the cleavage of the L-fucose moieties from glycostructures. They can perform fucosyl residue transfer reactions, known as transfucosylation, under appropriate reaction conditions, such as high substrate concentrations [20]. This unique ability of α-L-fucosidases has significant potential for synthesising fucosylated saccharides, major components of the oligosaccharides found in human milk.

According to the catalytic mechanism, α-L-fucosidases are divided into two subgroups: retaining α-L-fucosidases, which belong to the GH29 family, and inverting α-L-fucosidases, which belong to the GH95 family [21]. For transfucosylation to occur, the position of the anomeric carbon atom must be retained, which is why transfucosylation is characteristic for the GH29 family of fucosidases [22]. The retaining GH29 α-L-fucosidases employ a conventional Koshland double displacement mechanism that relies on two closely arranged carboxyl groups. First, one of the carboxyl groups serves as a general acid, facilitating the departure of the aglycone, while the second carboxyl group forms a covalent glycosyl-enzyme intermediate with an anomeric carbon atom. Second, the first carboxyl group functions as a general base, activating the fucosyl acceptor group for a nucleophilic attack on the anomeric carbon of the glycosyl-enzyme intermediate and causing transfucosylation to occure. The two-step process of two inversions gives the final product with a retained anomeric configuration identical to that of the starting substrate [23,24].

Microbiota from ruminants is an interesting and diverse source of enzymes such as glycoside hydrolases [25]. Alpacas are ruminant camelids whose faeces metagenome, as a source of fucosidases, has been scarcely investigated. In this study, data on four α-L-fucosidases from a metagenomic library of alpaca faeces are presented.

We have analysed the sequences and structures of these enzymes using bioinformatic methods. In addition, we have characterised the hydrolytic activity of the selected α-L-fucosidases. Finally, we evaluated the ability of the α-L-fucosidases to perform the transfucosylation reaction, which leads to oligosaccharide compounds like those found in human milk and fucosylated amino acids. Our results provide valuable insights into the use of α-L-fucosidases for synthesising fucosylated amino acids.

## 2. Results and Discussion

### 2.1. Isolation and Bioinformatic Analysis of α-L-Fucosidases

In this study, four clones, fuc25A, fuc25C, fuc25D, and fuc25E, capable of hydrolysing 5-bromo-4-chloro-3-indolyl-α-L-fucopyranoside were obtained from the metagenomic library of alpaca faeces. The plasmid DNA was extracted and analysed by sequencing. The plasmids of all selected clones contained fragments of 2.5–3.5 kb, and genes encoding proteins with a conserved α-L-fucosidase domain were identified. The genes and proteins were named after the plasmids. The Blastp search tool revealed that the proteins Fuc25A, Fuc25C, Fuc25D, and Fuc25E are homologous to α-L-fucosidases. The majority of Fuc25A, Fuc25C, and Fuc25D homologs belong to bacteria from the class *Clostridia*, phylum *Firmicutes*, which is most abundant in alpaca faeces [26] Fuc25E is homologous to *Clostridia* or *Lentisphaeria* bacteria proteins. Analysis of the protein sequences (using the Uniprot peptide search tool) showed that none of the proteins had a secretory signal peptide. The Fuc25A, Fuc25D, and Fuc25E proteins have comparably similar sizes. Their peptide chain consists of 425, 428, and 428 amino acids. Fuc25C is a slightly larger protein and consists of 499 amino acids. The amino acid identity values of Fuc25A, Fuc25D, and Fuc25E ranged from 49.15 to 53.94%. Fuc25C was the most dissimilar to other proteins examined, with identity values of less than 30% (Appendix A).

The phylogenetic analysis revealed that fucosidases Fuc25A, Fuc25D, and Fuc25E are closely related to each other and, together with protein from *Lacticaseibacillus paracasei*, they form a distinct branch on the phylogenetic tree (Figure 1). The latter protein has been characterised as α-L-fucosidase [27]. Another phylogenetic tree branch that is also more closely related to Fuc25A, Fuc25D, and Fuc25E contains protein sequences from *Bacteroides thetaiotaomicron* VPI-5482 (BT_2970) [28] and *Bacteroides fragilis* NCTC 9343 (BF0028) [17]. These sequences were also characterised as α-L-fucosidases. Even though the indicated proteins are α-L-fucosidases, significant differences between sequence identities are observed. Fuc25A, Fuc25D, and Fuc25E sequence identities with *L. paracasei* α-L-fucosidase are 45–49% and with *Bacteroides* spp. proteins BT_2970 and BF_0028 in the range of 24–27%. Although Fuc25A, Fuc25D, and Fuc25E have high sequence identity values compared to each other, they can be considered as distinct from other characterised α-L-fucosidases, as the sequence identity of most related characterised fucosidases does not reach 50%. 

The GH29 family is divided into GH29A and GH29B subfamilies depending on sequence homology, determining their substrate specificity [18,21]. α-L-Fucosidases of the GH29A subfamily has a broader substrate specificity compared to that of GH29B, and acts on α(1,2)-, α(1,3)-, α(1,4)-, and α(1,6)-fucosyl linkages. Members of the GH29B subfamily act on α(1,3)- and α(1,4)-linkages with a branched Gal only. GH29A fucosidases are active towards synthetic aryl substrates, e.g., *p*-nitrophenyl-α-L-fucopyranoside (*p*NP-αFuc), while GH29B fucosidases are inactive towards these substrates.

Determining the family of glycoside hydrolases helps to propose the reaction mechanism and substrate recognition patterns. The fucosidases from *L. paracasei* and *Bacteroides* spp. are assigned to the GH29A subfamily, which describes enzymes with broad substrate specificity. These enzymes are usually capable of the hydrolysing synthetic substrate, *p*NP-αFuc, and are assigned to the EC 3.2.1.51 [17,29]. The analysis using the dbCAN3 algorithm predicted that α-L-fucosidases from alpaca faeces metagenome, Fuc25A, Fuc25D, and Fuc25E, also belong to EC 3.2.1.51 and the GH29 family. Therefore, a retaining mechanism of action and broad substrate specificity of these fucosidases could be expected [17,29].

Fuc25C was found on a separate branch of the tree of α-L-fucosidases. For Fuc25C, the closest neighbours are α-L-fucosidases from *Ruminococcus* sp. SR1/5, *Zobellia galactanivorans*, and *Saccharolobus solfataricus* P2. These proteins were assigned to the GH29A subfamily and EC 3.2.1.51 [27,30]. Identities of Fuc25C with α-L-fucosidases from *Ruminococcus* sp. SR1/5, *Z. galactanivorans*, and *S. solfataricus* P2 are 34%, 31%, and 30%, respectively.

HHpred search revealed that the closest structural homologue for Fuc25A, Fuc25D, and Fuc25E is an α-L-fucosidase α-L-f1wt from *Paenibacillus thiaminolyticus* (6gn6, E-values: Fuc25A = 3.5 × 10^−72^, Fuc25D = 7.3 × 10^−73^, Fuc25E = 8.7 × 10^−72^). For Fuc25C, the closest structural homologue was identified as the α-L-fucosidase B5CYA5 from *Phocaeicola plebeius* DSM 17135 (7LJJ_A, E-value = 4.9 × 10^−79^) [31] and a significant similarity was also observed with α-L-f1wt from *P. thiaminolyticus* (6gn6, E-value = 2.2 × 10^−67^). Since catalytic amino acids are identified for α-L-f1wt, its structure was chosen for comparison with Fuc25A, Fuc25C, Fuc25D, and Fuc25E structure models generated by AlphaFold2. Additionally, template-based docking was performed using D-glucose from α-L-fucosidase α-L-f1wt.

To identify differences in the structures of the target α-L-fucosidases, the structure models of Fuc25A, Fuc25C, Fuc25D, and Fuc25E were compared. Overlay of the structure models showed that Fuc25C has additional unstructured loops and α-helixes compared to Fuc25A, Fuc25D, and Fuc25E (Figure 2).

The comparison of the Fuc25A, Fuc25D, and Fuc25E structure models showed high overall structure similarity. However, Fuc25E has a slightly different structural residue arrangement near the active site (Figure 3).

The active site amino acids of bacterial fucosidases are highly conserved. The catalytic amino acids of α-L-fucosidase α-L-f1wt from *P. thiaminolyticus* are D186 and E239 [32]. Thus, the catalytic amino acids identified for the fucosidases from the alpaca faeces metagenome appear to be D169 and E220 for Fuc25A, D240 and E289 for Fuc25C, D175 and E226 for Fuc25D, and D174 and E226 for Fuc25E. Structural differences were observed between Fuc25A, Fuc25D, and Fuc25E compared to Fuc25C when structure model information was combined with multiple sequence alignment and dbCAN3 analysis results. Fuc25A, Fuc25D, and Fuc25E show high surface exposure of conservative amino acids in the active site pocket. On the other hand, surface visualisation of the Fuc25C structure shows that the conservative amino acids found in the active site are more covered by neighbouring residues when compared to Fuc25A, Fuc25D, and Fuc25E (Figure 4).

To obtain more information on the active site pocket physicochemical parameters, target fucosidases were analysed using ProteinPlus, a structure-based modelling support server [33]. The tools DoGSiteScorer [34] and PoseEdit [35] were used to evaluate the active site pocket physicochemical properties and interactions with D-glucose, which was docked into the structures of α-L-fucosidases using template-based docking (Figure 5). 

The DoGSiteScorer algorithm allowed us to estimate the surface area, volume, depth, hydrophobicity, and enclosure values of the pockets on the protein surface. Data analysis showed that Fuc25A has the lowest active site surface area, volume, and depth but the highest hydrophobicity score compared to other target α-L-fucosidases (Table 1).

Other parameters, such as pocket enclosure, show that the Fuc25C enzyme has the most enclosed active site pocket, with an enclosure score 2–3-fold higher than that of Fuc25A, Fuc25D, and Fuc25E. The enclosure score may be related to substrate specificity, as the higher the enclosure score of the enzyme’s active site pocket, the more difficult it is for substrates to enter the pocket.

### 2.2. Gene Expression, Protein Synthesis, and Purification of α-L-Fucosidases

*Escherichia coli* BL21(DE3) was used for the overexpression of the α-L-fucosidases Fuc25A, Fuc25D, and Fuc25E, and *Escherichia coli* HMS174 (DE3) for Fuc25C. All enzymes were expressed with a C-terminal His-Tag from the pLATE31 expression vector and were produced in *E. coli* strains in a soluble state. Mannitol was added to improve the solubility of Fuc25C and Fuc25D proteins [36]. The target proteins were purified using a HiTrap Chelating HP (Ni-NTA Sepharose) affinity chromatography column. In the SDS-PAGE analysis (Figure 6), the purified proteins Fuc25A, Fuc25D, and Fuc25E were presented as nearly single bands in the range of 45–50 kDa, whereas Fuc25C was approximately 60 kDa. In all cases, the size of the protein bands was consistent with the theoretical molecular weight. Biochemical characterisation of the α-L-fucosidases was performed with the purified enzymes.

### 2.3. Characterisations of α-L-Fucosidases for Hydrolytic Activity

The artificial chromogenic substrate *p*NP-αFuc was used to determine the hydrolytic activity of the purified recombinant α-L-fucosidases. All four purified recombinant α-L-fucosidases from the alpaca faeces metagenome could hydrolyse *p*NP-αFuc. The activity of α-L-fucosidases was initially investigated in the pH range from 3.0 to 10. As all fucosidases (Fuc25A, Fuc25C, Fuc25D, and Fuc25E) were found to be more active at neutral pH (data not shown) than at acidic or alkaline pH, activity was further assayed at pH 6.0–8.0 (Figure 7a). We determined that all the fucosidases investigated were the most active at pH 7.0. In addition, they showed an activity of 50% or higher at the pH range from 6.0 to 8.0 (Figure 7a). Overall, the previously described fucosidases are active over a wide pH range (pH 3–9). Plant fucosidases prefer acidic pH [21]. The majority of bacterial fucosidases show a preference for neutral pH, while others favour slightly acidic or slightly alkaline conditions. The marine fucosidases FucWf1, FucWf2, and FucWf3 were most active at pH 6.3 [37]. The activities of soil fucosidases EntFuc, Mfuc1-5, and Mfuc7 were the highest at pHs 6 or 7 [24,38], while the fucosidase Mfuc6 was the most active at pH 9 [24]. The highest activity in an alkaline environment (pH 9) was determined for fucosidase rTfFuc1 from the periodontal pathogen *Tannerella forsythia* [39]. The results obtained in our study are consistent with reported data that gut microbiota fucosidases are most active at pH 6–8 [21].

We investigated the effect of pH and storage time on the stability of fucosidases (Figure 7b). Enzymes Fuc25A, Fuc25C, and Fuc25E were most stable at pH 8.0, as they retained 79, 87, and 75% of activity, respectively, even after preincubation for 24 h. Fuc25D was most stable when preincubated at pH 7.0, retaining 86% of its activity after 24 h. All enzymes were unstable at pH 3.0 and pH 10. Preincubation at pH 4.0 and 5.0 also contributed to the significant activity loss for each enzyme. To summarise, α-L-fucosidases investigated were stable at pH 6.0–9.0. The previously described enzymes of the GH29 family are stable in the pH range from 4 to 11: Afc2 at pH 5–11 [40], EntFuc at pH 5–8 [38], and BT_2970 and BT_2192 at pH 4–9 [28].

The thermal activity of the investigated fucosidases was somewhat different (Figure 8a). Fuc25A and Fuc25E preferred a 37 °C temperature, which is inherent to gut fucosidases [21]: BT_2970 from *B. thetaiotaomicron* [28] and fucosidases from *Bifidobacterium longum* subsp. *infantis* ATCC 15697 [41]. However, fucosidases from *Lactobacillus casei*, a bacterium commonly found in the human gut, were most active at 39 °C (AlfA) and 41 °C (AlfB and AlfC) [42]. Yet, the highest activity for the fucosidases Fuc25C and Fuc25D was at lower temperatures, 20 °C and 30 °C, respectively. Nevertheless, Fuc25D retained 97% activity at 37 °C temperature. The relative activity of Fuc25C was higher than 75% at 15 °C and 25 °C. Lower temperatures are more specific for marine fucosidases, e.g., FucWf1, FucWf2, and FucWf3 from *Wenyingzhuangia fucanilytica* [37]. A soil-oriented fucosidase EntFuc isolated from *Enterococcus gallinarum* showed the highest hydrolytic activity at 30 °C [38]. Fucosidases Fuc25A and Fuc25C were completely inactive at 55 °C and 45 °C, respectively, while Fuc25D and Fuc25E were at 50 °C. It can be summarised that fucosidases Fuc25A, Fuc25D, and Fuc25E probably originate from intestinal bacteria, as their peak activity temperature is close to the rectal temperature of the alpaca, 38.5–39.5 °C [43]. In contrast, Fuc25C is most likely from soil bacteria as it was most active at temperatures typical for environmental fucosidases.

The fucosidases investigated were most stable at 0 °C. They retained more than 80% of activity within 24 h of storage (Figure 8b). At higher temperatures, the stability of the fucosidases decreased depending on the preincubation temperature and time. Still, all fucosidases retained about 40% of activity after preincubation at 20 °C (room temperature) for 24 h. In all cases, the residual activity was about 60% after preincubation for 1 h at 20 °C or 30 °C. The stabilities of Fuc25A and Fuc25E were poor at their highest activity temperature (37 °C). Their residual activities were less than 20% after 24 h. However, Fuc25C retained 50% of its activity after 24 h of storage at 20 °C. Therefore, it is more stable at its optimum activity temperature than the previously mentioned proteins. The temperature of 45 °C was critical to all enzymes, especially Fuc25C. All obtained results of Fuc25A, Fuc25C, Fuc25D, and Fuc25E are consistent with previously reported findings that gut, marine, and soil fucosidases are stable at temperatures below 37 °C and lose their activities at 45 °C or higher temperatures [24,28,37,38].

The kinetic parameters of all α-L-fucosidases investigated were evaluated against *p*NP-αFuc at pH 7.5 and room temperature. The lowest K_M_ value was for Fuc25D (79.7 ± 8.6 µM) and the highest for Fuc25C (1401.1 ± 373.6 µM) (Table 2). The k_cat_ value of Fuc25D was calculated to be 28.87 ± 1.56 s^–1^, and it was the highest of all enzymes studied. Consequently, Fuc25D had the highest substrate affinity and catalytic efficiency of the fucosidases studied. The catalytic efficiencies of Fuc25A and Fuc25E were one order of magnitude lower than those of Fuc25D. The values of all kinetic parameters calculated for the enzymes Fuc25A, Fuc25C, Fuc25D, and Fuc25E fall within the range of relevant values determined for bacterial fucosidases (K_M_ µM to mM, k_cat_ from 10^–3^ to 10^–2^ s^–1^, k_cat_/K_M_ from 10^–6^ to 10^–2^ µM^–1^ s^–1^ [21]). Fuc25D (K_M_ 79.1 ± 8.6 µM) and Fuc25E (K_M_ 85.1 ± 12.9 µM) have the highest substrate affinity of previously described gut fucosidases except SrFucNaFLD (K_M_ 10.59 ± 2.64 µM) [44]. Although Fuc25C has the lowest affinity among Fuc25 fucosidases, its affinity is higher than AlfB (K_M_ 2900 µM) and AlfC (K_M_ 5200 µM) fucosidases [42]. Fuc25D k_cat_ value is one of the highest among described gut fucosidases. Greater values were determined only for Amuc_0010 (378.33 s^–1^) from *Akkermansia muciniphila* MucT (ATCC BAA-835) [45] and ATCC_03833 (83.6 ± 2.97 s^–1^) from *Ruminococcus gnavus* ATCC 29149 [46].

In this study, 2′-fucosyllactose and 3-fucosyllactose were subjected to hydrolysis with Fuc25A, Fuc25C, Fuc25D, and Fuc25E enzymes. All tested fucosidases were capable of hydrolysing 2′-fucosyllactose. Based on substrate spot decay and the appearance of lactose and fucose products, the most efficient 2′-fucosyllactose was hydrolysed by Fuc25D and Fuc25E (Figure 9). None of the fucosidases investigated were capable of hydrolysing 3-fucosyllactose, no formation of lactose or fucose was observed in TLC. Thus, Fuc25A, Fuc25C, Fuc25D, and Fuc25E are regioselective to 2′-fucosyllactose.

### 2.4. Transfucosylation Catalysed by Fuc25A, Fuc25C, Fuc25D, and Fuc25E

α-L-Fucosidases which catalyse transfucosylation provides an alternative pathway to produce human milk components [22]. The capability of α-L-fucosidases from the metagenome of alpaca faeces to perform transfucolylation was investigated using HPLC-MS. After incubation with Fuc25A, Fuc25C, Fuc25D, and Fuc25E, reaction mixtures were screened for molecular ions that could indicate transfructosylated acceptor substrate analogue. Mass scans were compared with those of negative control samples.

Initially, the reaction was performed with mono-, di-, and oligosaccharides as fucosyl group acceptors, while *p*NP-αFuc acted as a donor of fucosyl residues. The transfucosylation results are presented in Table 3 and Appendix A. All four α-L-fucosidases were able to transfucosylate monosaccharides. Pentose sugars, such as D-xylose and D-ribose, were shown to be suitable acceptors for the fucosyl group transfer. The reaction was observed after indicating the presence of reaction products ([M+K^+^]^+^ = 335.80 Da for D-xylose; [M+K^+^]^+^ = 334.90 Da for D-ribose) (Appendix A).

Deoxy-hexose sugars such as L-fucose and L-rhamnose were also suitable acceptors for the transfucosylation reaction. Molecular ions of L-fucose and L-rhamnose transfucosylated analogues were observed after incubation with Fuc25A, Fuc25C, Fuc25D, and Fuc25E (([M+K^+^]^+^ = 348.90 Da for L-fucose; [M+K^+^]^+^ = 349.00 Da for L-rhamnose) (Appendix A). However, the L-fucose transfucosylation reaction with Fuc25A, Fuc25D, and Fuc25C produced only traces of fucosylated fucose analogues, whereas the reaction with L-rhamnose gave much better results (Appendix A). Poor transfucosylation reaction results with L-fucose as an acceptor may be due to the inhibition of α-L-fucosidases by the substrate [47].

Hexose sugars such as D-glucose, D-galactose, D-fructose, and D-mannose were also suitable fucosyl acceptors for all α-L-fucosidases investigated. Molecular ions that indicate the formation of transfucosylated donor substrate were identified in the reaction mixture after incubation with fucosidases (Appendix A). In addition, all target enzymes could perform transfucosylation reactions with tested D-sugar isomers and L-fucose and L-rhamnose, which are L-sugar isomers. Monosaccharides such as D-glucose and D-galactose are found in human milk as significant components of HMOs. Therefore, the ability to generate fucosylated saccharide analogues shows the potential to create compounds like those found in HMOs. Other researchers employ α-L-fucosidases for the synthesis of various fucosylated glycosides, such as fucosyllactose [24], lacto-*N*-fucopentaose [22], and their analogues [18,48]. BF0028 α-L-fucosidase from *B. fragilis*, which is related to Fuc25A, Fuc25D, and Fuc25E (Figure 1), could not perform transfucosylation reaction with glucose, maltose, lactose, galactose, and fructose as fucosyl acceptors. In contrast, the *B. fragilis* α-L-fucosidase BF3242, which is less like the fucosidases we characterised, was able to transfer the fucosyl group onto maltose and galactose [17].

The successful transfucosylation of hexoses encouraged us to test more complex compounds, such as an amide derivative of glucose, *N*-acetylglucosamine. *N*-Acetylglucosamine fucosylated products were described in other studies [17,49]. The phylogenetically related α-L-fucosidase BF0028 from *B. fragilis* was assessed as a potential catalyst for tranfucosylation of *N*-acetylglucosamine. However, BF0028 did not show the ability to produce fucosylated *N*-acetylglucosamine. Still, α-L-fucosidase BF3242 from *B. fragilis* found on a more distant tree branch showed the ability to produce fucosylated *N*-acetylglucosamine [17]. We synthesised fucosyl-*N*-acetylglucosamine disaccharides using 0.5 M of *N*-acetylglucosamine as a fucosyl group acceptor and 20 mM of *p*NP-αFuc as a fucosyl donor. The reaction with the target fucosidases was performed at a pH of 7.5 and a temperature of 25 °C. *N*-Acetylglucosamine was fucosylated by Fuc25A, Fuc25C, Fuc25D, and Fuc25E. Hence, Fuc25A, Fuc25D, and Fuc25E displayed a broader acceptor scope than their related sequence of α-L-fucosidase BF0028 [17].

Finally, all α-L-fucosidases were subjected to transfucosylation of amino acids. Both isomers (D- and L-) of serine and threonine were used as fucosyl acceptors. HPLC-MS results showed a new compound formation with a mass that could be assigned to a fucosylated amino acid ([M+K^+^]^+^ = 290 Da for D- and L- serine; [M+K^+^]^+^ = 304 Da for D- and L- threonine) (Appendix A). In the negative control sample, mass peaks that could be assigned to fucoylated amino acids were not observed.

α-L-Fucosidases were capable of transfucosylating both D- and L-isomers of serine and threonine (Table 3); however, stronger product signals were detected when reactions were performed with D-amino acids (Appendix A). Although the transfucosylation reaction with *N*-(tert-butoxycarbonyl)-L-serine methyl ester and *N*-(tert-butoxycarbonyl)-L-threonine methyl ester has been described [50] and the metabolic pathway that utilises the glyco amino acid, fucosyl-α-1,6-*N*-GlcNAc-Asn, has been identified [51], the transfucosylation of a single amino acid has not been observed to date. The ability of α-L-fucosidases to perform amino acid transfucosylation could be beneficial for the synthesis of fucose-amino acid conjugates or glycopeptides.

To compare transfucosylation efficiency among the fucosidases studied, the reactions were performed using normalised protein concentrations of 0.02 mg/mL. The reactions were performed in potassium phosphate buffer with D-glucose, D-xylose, D-trehalose, D-raffinose, maltotriose, or oligosaccharide mixtures (galactooligosaccharides and xylooligosaccharides) used as fucosyl acceptors. According to HPLC-MS data, Fuc25A generated the highest amounts of fucosylated products of D-glucose, D-xylose, D-trehalose, and D-raffinose (Appendix A). These results align with the physicochemical parameters calculated for Fuc25A since it had the highest hydrophobicity and the lowest enclosure score. High active site hydrophobicity could also contribute to better fucosyl functional group donor hydrophobic substrate *p*NP-αFuc binding and high V_max_ value when *p*NP-αFuc is used for enzyme activity measurements [52]. In addition, a higher amount of hydrogen bonds could contribute to different substrate orientations in the active site (Appendix A). The lowest amount of fucosylated products was observed after catalysed reactions by Fuc25C. The reaction with Fuc25E generated the highest amount of fucosylated maltotriose products, while Fuc25C produced the lowest amount. All fucosidases investigated were used to perform transfucosylation reactions with xylooligosaccharides and galactooligosaccharides as fucosyl acceptors. Xylooligosaccharides seemed to be better fucosyl group acceptors than galactooligosaccharides because only traces of fucosylated products were observed in reactions where galactooligosaccharides were used. Xylooligosaccharides with a polymerisation degree of 2 (DP2) and 3 (DP3) were transfucosylated by Fuc25A, Fuc25D, and Fuc25E. In addition, Fuc25A and Fuc25E could produce fucosylated DP4 xylooligosaccharide (Appendix A). Xylooligosaccharides and galactooligosaccharides are known to have prebiotic effects on human health [53,54]. Transfucosylation of these compounds could generate various oligosaccharides that might provide additional health benefits and even mimic the benefits of human milk oligosaccharides [55]. The transfucosylation position of compounds synthesised by target fucosidases is unknown and will be the subject of further research. The results revealed that even homologous α-L-fucosidases belonging to the same GH29A subfamily can differ significantly in their conditions of action, substrate specificity, etc., and that, despite the previously described wide diversity of α-L-fucosidases, the isolation and characterisation of novel fucosidases is still needed.

This study evaluated Fuc25A, Fuc25C, Fuc25D, and Fuc25E structural differences, active site pocket parameters, and enzyme hydrolytic and transfucosylation capabilities. Fuc25C fucosidase stood out with higher enclosure values and lower hydrophobicity values. We also noticed that Fuc25C has higher K_M_ values, showing lower affinity to the substrate *p*NP-αFuc. Hydrophobicity values of active site regions could be related to this substrate recognition. We also observed that Fuc25C transfucosylation abilities under the conditions tested were the lowest. This enzyme produced less transfucosylated products than Fuc25A, Fuc25D, and Fuc25E. This could also be related to the lower hydrophobicity of the active site region since the fucosyl donor substrate used for transfucosylation reactions was *p*NP-αFuc.

Moreover, higher enclosure values of this active site pocket in Fuc25C could impact the ability to accept various acceptor substrates since their structures and sizes can differ, starting from monosaccharides and finishing with amino acids or branched oligosaccharides. Finally, Fuc25C had the most structural differences compared to the Fuc25A, Fuc25D, and Fuc25E. Unstructured parts of Fuc25C (Figure 2, region 3) were observed near the active site, which could affect substrate recognition as they can be flexible. Conformational changes of unstructured areas and their importance in substrate recognition were observed in α-L-fucosidase AlfC from *L. casei* [56]. Nevertheless, more data are needed on the characteristics of the produced transfucosylated compounds. Also, experimentally determining α-L-fucosidases structures could provide more insights into catalytic reaction mechanisms and factors that limit transfucosylation capabilities.

## 3. Materials and Methods

### 3.1. Materials

*E. coli* DH5α and the pUC19/HindIII vector were used to construct and screen the metagenomic library. *E. coli* BL21(DE3) and *E. coli* HMS174 (DE3) with the pLATE31 vector used for gene expression. *E. coli* BL21(DE3) was used as a host for the synthesis of recombinant α-L-fucosidases Fuc25A, Fuc25D, and Fuc25E, whereas *E. coli* HMS174(DE3) was used as a host for α-L-fucosidase Fuc25C. ZR Quick-DNA Fecal/Soil Microbe MiniPrep Kit was obtained from Zymo Research ((Irvine, CA, USA). pUC19/HindIII, isopropyl-β-D-thiogalactopyranoside (IPTG), pLATE31 cloning kit was obtained from Thermo Scientific (Vilnius, Lithuania). Tris-HCl, Tris Base, *p*-anisaldehyde, D-raffinose, D-glucose, L-rhamnose, D-mannose, D-maltose, maltotriose, D-threonine, D-serine, L-threonine, PMSF, and imidazole were purchased from Sigma-Aldrich (St. Louis, MO, USA). TLC plates, D-fructose, D-ribose, acetonitrile, and L-serine were purchased from Merck (Rahway, NJ, USA). 5-Bromo-4-chloro-3-indolyl-α-L-fucopyranoside, L-fucose, *p*NP-αFuc, 2′-fucosyllactose, and 3-fucosyllactose were purchased from Biosynth (Staad, Switzerland). GOS were obtained from Pienas LT (Kaunas, Lithuania). Formic acid was purchased from Reaxim. D-Trehalose was purchased from PanReac AppliChem (Darmstadt, Germany). XOS, D-galactose, D-xylose, lactose, and dipotassium phosphate were purchased from Roth (Karlsruhe, Germany). Potassium dihydrogen phosphate was purchased from Fluka (Charlotte, NC, USA). *N*-Acetylglucosamine was purchased from Acros Organics (Geel, Belgium).

### 3.2. DNA Extraction, Construction, Screening of Metagenomic Library, and DNA Sequencing

DNA was extracted from alpaca faeces using the ZR Quick-DNA Fecal/Soil Microbe MiniPrep Kit to construct the metagenomic library. The total DNA was partially hydrolysed using restriction endonuclease HindIII. The obtained fragments were cloned into the pUC19 vector and used to transform *E. coli* DH5α competent cells by electroporation as described [57]. The size of inserted fragments and the quality of the metagenomic library were estimated as described previously [58].

The functional screening of fucosidases was performed using LB agar plates supplemented with 3 mM 5-bromo-4-chloro-3-indolyl-α-L-fucopyranoside, 1 mM IPTG, and 100 µg/mL of ampicillin. Recombinant bacteria harbouring fucosidase formed blue/dark colour colonies.

Nucleotide sequences of recombinant plasmids from selected clones were determined at MacrogenEurope (Amsterdam, Netherlands) using the following sequencing primers: M13F-pUC (5′-GTTTTCCCAGTCACGAC-3′), M13R-pUC (5′-CAGGAAACAGCTATGAC-3′). Sequences were analysed using the VectorNTI program version 11.5.1 (Invitrogen, Waltham, MA, USA), and a homology search was conducted using the Blast server (http://www.ncbi.nlm.nih.gov/BLAST, accessed on 5 September 2023).

Metagenome-derived fragments containing genes of α-L-fucosidases were deposited to GenBank and obtained the following accession numbers: OR873635 for pfuc25ApUC, OR873636 for pfuc25CpUC, OR873637 for pfuc25DpUC, and OR873638 for pfuc25EpUC. The accession numbers of the proteins are as follows: WPO27445 for Fuc25A, WPO27446 for Fuc25C, WPO27447 for Fuc25D, and WPO27449 for Fuc25E.

### 3.3. Sequence, Phylogeny, and Structure Model Analysis

Amino acid sequences of α-L-fucosidases were used for Blast search using default parameters and performing search on non-redundant sequences database. The Clustal Omega multiple sequence alignment tool [59] assessed the identity between the analysed α-L-fucosidases. Default parameters of the Clustal Omega algorithm were used. The dbCAN3 tool [60], which evaluated fucosidases in the CAZy (Carbohydrate Active Enzymes database (http://www.cazy.org/, accessed on 2 June 2023) [61]), was used to predict glycan substrates and assign fucosidase families. Default search parameters were used with tools HMMER: dbCAN, DIAMOND: CAZy, and HMMER: dbCAN-sub.

The phylogenetic tree was built using CD-HIT (v4.8.1) [62]. The Mafft (v7.515) [63] algorithm, dedicated to multiple sequence alignment (MSA), was used for sequence clustering. The alignement was trimmed to remove phylogenetically uninformative sites using the Clipkit tool (v1.3.0) and subjected to evolutionary model evaluation using modeltest-ng [64]. The evolutionary model found to be the best (LG+I+G4+F) for assessing the phylogeny of the sequences was used to build the phylogenetic tree raxml-ng (v0.9.0) [65]. Multiple sequence alignment preparation was performed in Jupyter Lab (v3.0.14), and determination of the evolutionary model and construction of the phylogenetic tree was carried out in a Google Collaboratory environment. The method for generating a phylogenetic tree was described previously [66]. The phylogenetic tree was visualised using iTOL [67]. dbCAN3 and CAZy data were used for annotation [62,63,64,65,66,67].

HHpred from Toolkit Toobingen [68,69] determined the closest homologues in the PDB database. The search was carried out using default parameters and database version PDB_mmCIF_70_18_Jun. ColabFold (v1.5.2) was used for structure model determination. All parameters were at their default values, the number of recycles was increased to 12, and amber relaxation was performed for each structure evaluated as top-ranked.

Template-based docking with D-glucose was performed using the structure of α-L-fucosidase α-L-f1wt from *Paenibacillus thiaminolyticus* (6gn6). Active site pocket and D-glucose binding were evaluated using ProteinPlus [33] server tools DoGSiteScorer [34] and PoseEdit [35]. Active site pocket parameters (surface, volume, depth, hydrophobicity, enclosure) were determined by DoGSiteScorer. PoseEdit assessed D-glucose interactions with active site pocket amino acid residues.

Structure comparison and visualisation were performed using UCSF Chimera software (v1.15) [70]. Non-structured N and C end parts in Figure 2, Figure 3 and Figure 4 were hidden for visualisation purposes and did not affect the main message provided in the text. These parts of structures were visualised as Fuc25A (8–417 aa), Fuc25D (17–420 aa), Fuc25C (1–491 aa), and Fuc25E (7–420 aa).

The default parameters of the Mafft algorithm generated an alignment of GH29 members from the CAZy database and Fuc25A, Fuc25D, Fuc25C, and Fuc25E to identify highly conservative amino acids. Jalview [71] was used to visualise the MSA data for structure visualisation.

### 3.4. Overexpression and Purification of Enzymes

Metagenome-derived genes of α-L-fucosidases (Fuc25A, Fuc25C, Fuc25D, and Fuc25E) were amplified using specific pLATE31 vector-compatible primers (Appendix A) and Phusion Plus DNA Master Mix. Obtained fragments were cloned into the pLATE31 vector. Further on, the constructed plasmids were transformed into *E. coli* BL21 (DE3) (Fuc25A, Fuc25D, Fuc25E) and *E. coli* HMS174 (DE3) (Fuc25C) for protein overexpression. The recombinant bacteria plasmids were cultured in LB medium containing 100 μg/mL of ampicillin by shaking at 37 °C until the optical density at 600 nm reached 0.6–0.8, then IPTG was added to the final concentration of 0.1 mM. In the case of Fuc25C and Fuc25D, the medium was also supplemented with 5% D-mannitol. The bacteria were grown overnight (20 h) aerobically at 20 °C. Subsequently, the cells were harvested by centrifugation and resuspended in a 50 mM potassium phosphate (pH 8) buffer containing 300 mM KCl, 10% glycerol, and 1 mM PMSF. Then, cells were lysed by ultrasonication (30-s impulse, with 10-s pauses every 5 s). The debris was collected by centrifugation at 15,000× *g* for 10 min at 4 °C. Cleared lysate was applied to the nickel affinity chromatography column HiTrap Chelating HP (Cytiva, Marlborough, MA, USA). The column was equilibrated and washed with buffer A (50 mM potassium phosphate buffer, pH 8, containing 300 mM potassium chloride). Fuc25A, Fuc25D, and Fuc25E were eluted using buffer A supplemented with 500 mM imidazole, while Fuc25C with buffer A containing 350 mM imidazole. An elution gradient length of 20 column volumes was used. Eluted fractions were pooled, and imidazole was removed using a HiTrap Desalting column (Cytiva, Marlborough, MA, USA). After desalting, the proteins were concentrated by centrifugation using 30 kDa Spin-X UF filter columns (Corning, New York, NY, USA). The SDS-PAGE gel (5% concentrating and 14% resolving) was used to analyse the purified samples. The gel was stained with Coomassie Brilliant Blue R-250 and washed with a 7% acetic acid solution. The concentrations of the purified protein solutions were determined using the Folin–Ciocalteau reagent [72].

### 3.5. Characterisation of α-L-Fucosidases Hydrolytic Activity

The hydrolytic activity of α-L-fucosidases was investigated using *p*NP-αFuc as a substrate and measuring the amount of *p*-nitrophenol at 410 nm spectrophotometrically. The reaction mixture (1 mL) contained 50 mM potassium phosphate buffer (pH 7.5) and substrate concentrations of 0.25 mM, 0.5 mM, 0.15 mM, and 0.3 mM for Fuc25A, Fuc25C, Fuc25D, and Fuc25E, respectively. The amounts of the enzymes used in the hydrolysis reaction were as follows: 4.15 μg of Fuc25A, 20.25 μg of Fuc25C, 0.83 μg of Fuc25D, and 0.92 μg of Fuc25E, unless otherwise stated. All reactions were performed at room temperature (20 °C) for 2 min. Measurements were repeated three times.

The optimal pH of α-L-fucosidases towards *p*NP-αFuc was analysed in the pH range 6–8 using 50 mM sodium citrate (pH 6, 6.5) and 50 mM sodium phosphate (pH 7, 7.5, 8) buffers. The pH stability of the enzymes was evaluated by incubating them in different buffers for 0.5, 1, 2, 6, and 24 h at 0 °C temperature, and the residual activity was measured. To determine the optimal activity temperature for the hydrolysis, the activity of each enzyme was measured at different temperatures from 10 to 55 °C. The thermostability was determined by evaluating the residual activity after incubating at various temperatures for 1, 6, and 24 h.

We considered the protein to be most stable in the environment where the relative activity change between the start and the endpoint was the lowest. The start point measurements were made after the protein incubation in a particular pH buffer or at the temperature tested for 30 s. The relative activity of each enzyme was calculated separately for every experiment. The highest value of activity obtained was set at 100%. The remaining values were standardised to the highest value.

To determine V_max_, K_M_, and k_cat_, eight concentrations of *p*NP-αFuc ranging from 0.02 to 2 mmol/L were used. The composition of the reaction mixture was as described above. The reaction was carried out at room temperature (20 °C) for 1 min, with signal recording every 1 s. Enzyme activity measurements were conducted five times for each substrate concentration. The activity of each enzyme was evaluated by calculating the amount of product (*p*-nitrophenol) formed from the measured absorbance of the reaction mixture at 410 nm. Product concentrations were computed using the extinction coefficient of *p*-nitrophenol at pH 7.5 (15.6 × 10^3^) [73]. V_0_ was determined from the change in product concentration between 10 and 40 s. The initial values of the reaction rates were obtained by fitting the Beer–Lambert law. The Michaelis–Menten kinetic model was used to calculate kinetic constants, and the analysis was performed using GraphPad Prism 9.5.1. [74].

### 3.6. Characterisation of the Transfucosylation of the Recombinant α-L-Fucosidases

The transfucosylation ability of α-L-fucosidases was determined using mono-, di-, and oligosaccharides and amino acids. Reactions were performed using 50 mM potassium phosphate buffer (pH 7.5). *p*NP-αFuc served as a fucosyl functional group donor with a final concentration of 20 mM. Fucosyl functional group acceptors were used at a final concentration of 0.3 M. The final concentration of each enzyme was 0.016 mg/mL for Fuc25A, 0.027 mg/mL for Fuc25C, 0.01 mg/mL for Fuc25D, and 0.026 mg/mL for Fuc25E, unless otherwise stated. Transfucosylation reactions with some of the acceptors (D-glucose, D-xylose, D-trehalose, raffinose and maltotriose, galactooligosaccharides, and xylooligosaccharides) were also performed using a normalised working protein concentration of 0.02 mg/mL.

Reactions were carried out for 2.5–4.5 h for mono-, di-, and oligosaccharides and 18 h for amino acids as acceptors. A temperature of 25 °C and agitation of 450 rpm were employed to support the enzymatic reaction. Reactions were stopped by mixing the sample with acetonitrile at the ratio of 1:1. Samples were analysed using HPLC-MS. Transfucosylation activity was evaluated using mass spectra peak area.

The peak area of a corresponding product mass was calculated via integration in the negative control sample (reaction mixture without protein) and in the samples after performing the reaction with α-L-fucosidases. The peak area of the transfucosylation reaction sample was divided by the negative control peak area. Suppose the difference (x) was 1 < x < 3; sample transfucosylation reaction product formation was observed at the Traces level (“Traces” in the table). If the difference was <1, the transfucosylation was considered not occurring (‘-” in the table). If the difference observed was >3, a transfucosylation reaction occurred (“+” in the table). Since reactions under normalised conditions were performed to compare enzymes’ transfucosylation abilities, the transfucosylation abilities were evaluated in percentages: the highest transfucosylated product area observed was evaluated as 100%. Peak areas generated by other fucosidases using the same fucosyl group acceptor were then respectfully considered in a percentage term.

### 3.7. Determination of the Hydrolysis of 2′-Fucosyllactose and 3-Fucosyllactose Catalysed by the Recombinant α-L-Fucosidases Using TLC

2′-Fucosyllactose or 3-fucosyllactose at the final concentration of 10 mM were used for the hydrolysis reaction (50 mM Tris-HCl buffer, pH 7.5). The reaction was carried out for ~29 h at 25 °C temperature using agitation of 450 rpm. Enzyme concentrations used for the hydrolysis reaction were 0.02 mg/mL. The reaction was stopped by a 1:1 dilution of the sample with acetonitrile. The samples were spotted (1–2 μL) onto TLC Silica Gel 60 F254 plates (Merck, Darmstadt, Germany). A TLC mobile phase of butanol: acetic acid: water in a volume ratio of 2:1:1 was used to separate 2′-fucosyllactose or 3-fucosyllactose. The plates were then dried and stained using *p*-anisaldehyde.

### 3.8. Determination of Transfucosylation Activity Using HPLC-MS

Transfucosylated compound identification was performed using a high-pressure liquid chromatography-mass spectrometry system (HPLC-MS, Shimadzu, Kyoto, Japan). A sample mixed with acetonitrile was centrifuged at 14,000× *g* for 10 min and injected into the system. Electrospray ionisation and time of flight detection are applied. The chromatographic separation was performed using the C18 column of 150 × 3 mm (YMC, Kyoto, Japan). Separation was carried out at 40 °C. The mobile phases used for the separation were water with 0.1% formic acid (solvent A) and acetonitrile (solvent B). Separation was performed using a combined isocratic and gradient elution method: 5% B for 1 min, from 5 to 95% B over 5 min, 95% B for 2 min, from 95 to 5% B over 1 min, and 5% B for 4 min. Flow rate 0.45 mL/min. For result analysis, LabSolutions LCMS software (v5.42 SP6) was used.

## 4. Conclusions

α-L-Fucosidases Fuc25A, Fuc25C, Fuc25D, and Fuc25E from alpaca faeces metagenome are mesophilic fucosidases belonging to the GH29A subfamily of hydrolases. All four enzymes hydrolyse 2′-fucosyllactose at different levels but are inactive towards 3-fucosyllactose. The investigated fucosidases can transfucosylate saccharides of varying chain lengths and unmodified amino acids harbouring hydroxy group. Overall, this study provides valuable information regarding the possible application of α-L-fucosidases in the synthesis of fucosylated amino acids. However, such application is currently limited, as the processes have low efficiency due to the hydrolytic activity. In the future, we plan to mutate α-L-fucosidases to obtain efficient α-L-fucosynthases. 

## Figures and Tables

**Figure 1 ijms-25-00809-f001:**
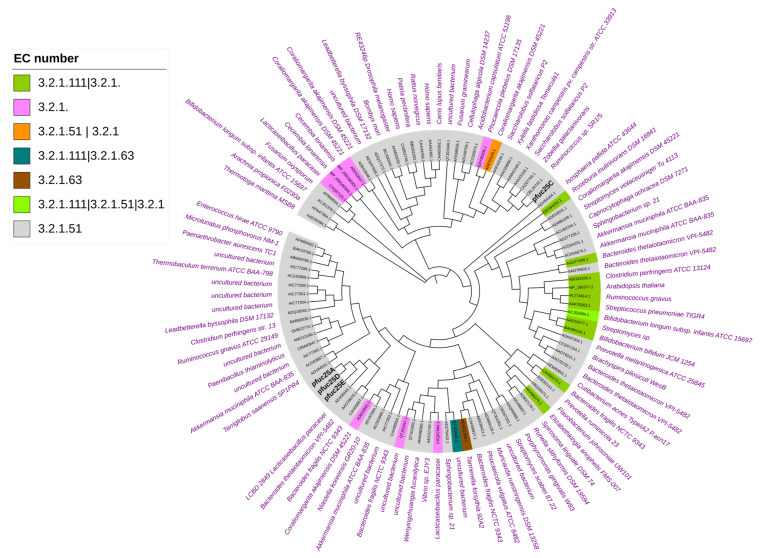
Phylogenetic tree of α-L-fucosidases from alpaca faeces metagenomic library. The maximum likelihood method was used for tree construction. The tree was constructed using characterised GH29 family members from the CAZy database. The legend provides information for the assigned EC numbers.

**Figure 2 ijms-25-00809-f002:**
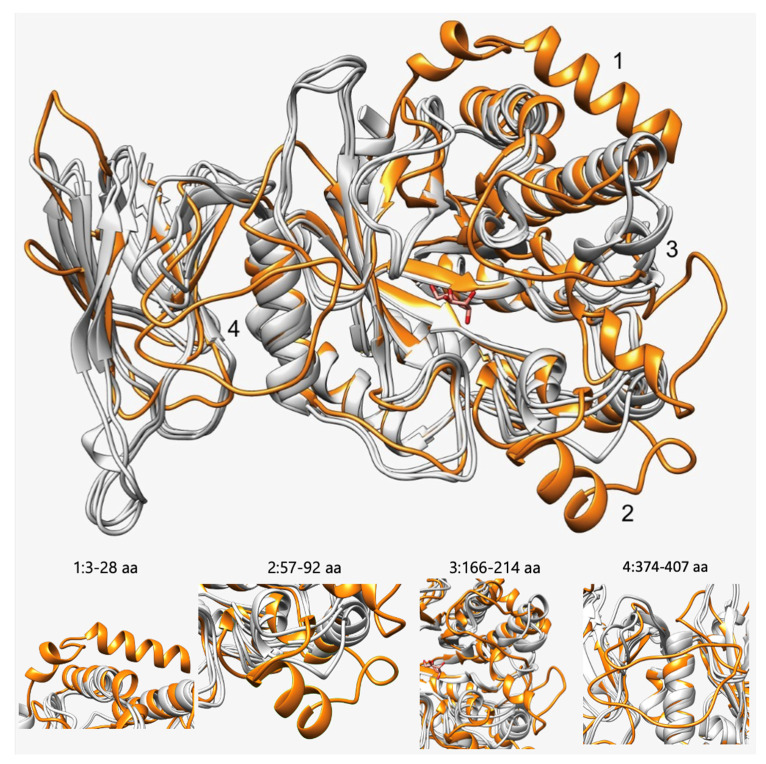
Overlay of Fuc25A, Fuc25C, Fuc25D, and Fuc25E structure models. Fuc25C—orange, Fuc25A, Fuc25D, and Fuc25E—grey. Numbers (1–4) are assigned to structural parts where differences are observed. Each part is shown in a zoomed-in view with the corresponding number and amino acid residues. D-glucose is shown in red.

**Figure 3 ijms-25-00809-f003:**
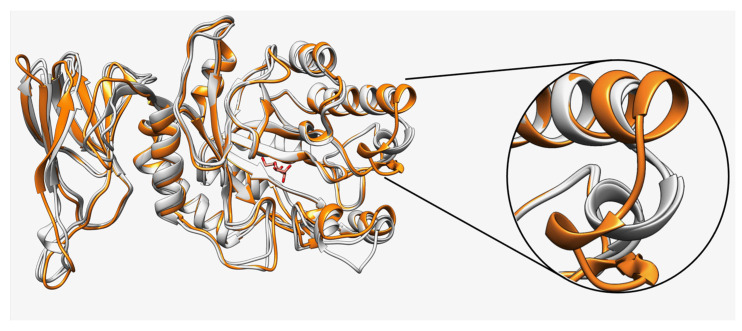
Overlay of Fuc25A, Fuc25D, and Fuc25E structure models. Fuc25E—orange, Fuc25A and Fuc25D—grey. Structural differences are shown in a zoomed-in view. D-Glucose is shown in red.

**Figure 4 ijms-25-00809-f004:**
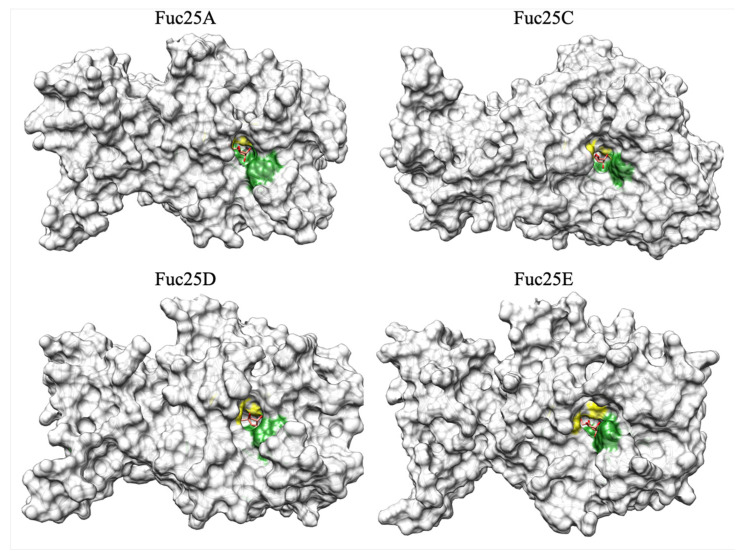
Conservative amino acids of the active site of Fuc25A, Fuc25C, Fuc25D, and Fuc25E and surface visualisation of the structure models. The green color shows highly conservative amino acids. Yellow–catalytic amino acids. D-Glucose is shown in red.

**Figure 5 ijms-25-00809-f005:**
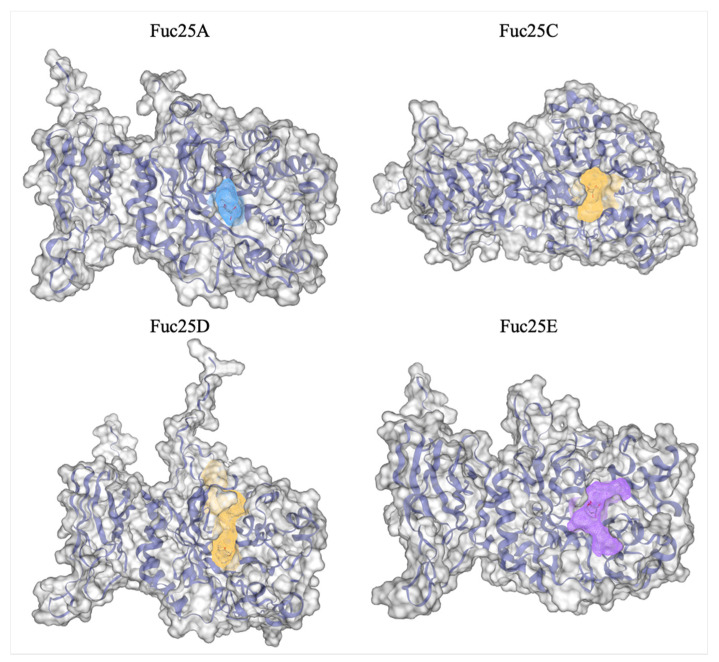
Fuc25A, Fuc25C, Fuc25D, and Fuc25E structure models with visualisation of the active site pockets. Pockets attributed to the active sites are calculated and evaluated by DoGSiteScorer. The structure within the active site pocket is D-glucose.

**Figure 6 ijms-25-00809-f006:**
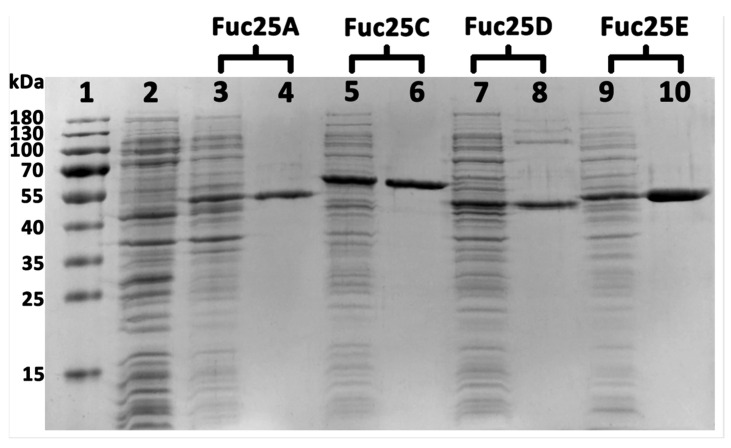
SDS-PAGE of the purified α-L-fucosidases. Lane 1—standard marker; lane 2—crude pET21b; lanes 3, 5, 7, 9—crude investigated fucosidases; lanes 4, 6, 8, 10—purified fucosidases by Ni-affinity chromatography.

**Figure 7 ijms-25-00809-f007:**
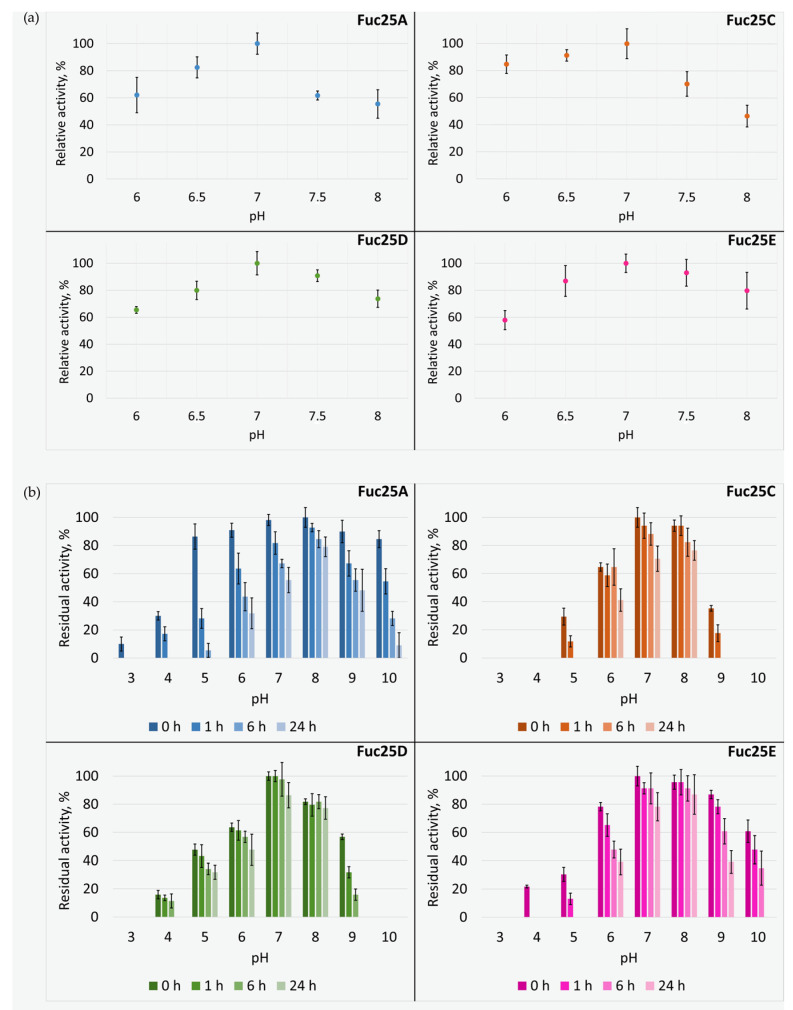
The effect of pH and storage time on the activity (**a**) and the stability (**b**) of α-L-fucosidases Fuc25A, Fuc25C, Fuc25D, and Fuc25E.

**Figure 8 ijms-25-00809-f008:**
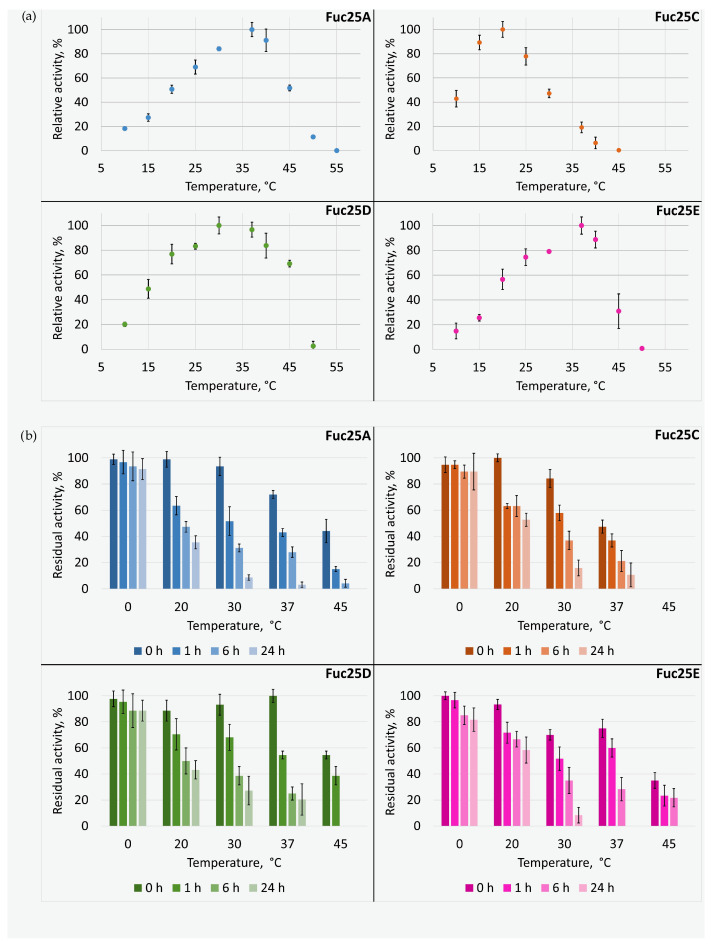
The effect of temperature and storage time on the activity (**a**) and the stability (**b**) of α-L-fucosidases Fuc25A, Fuc25C, Fuc25D, and Fuc25E.

**Figure 9 ijms-25-00809-f009:**
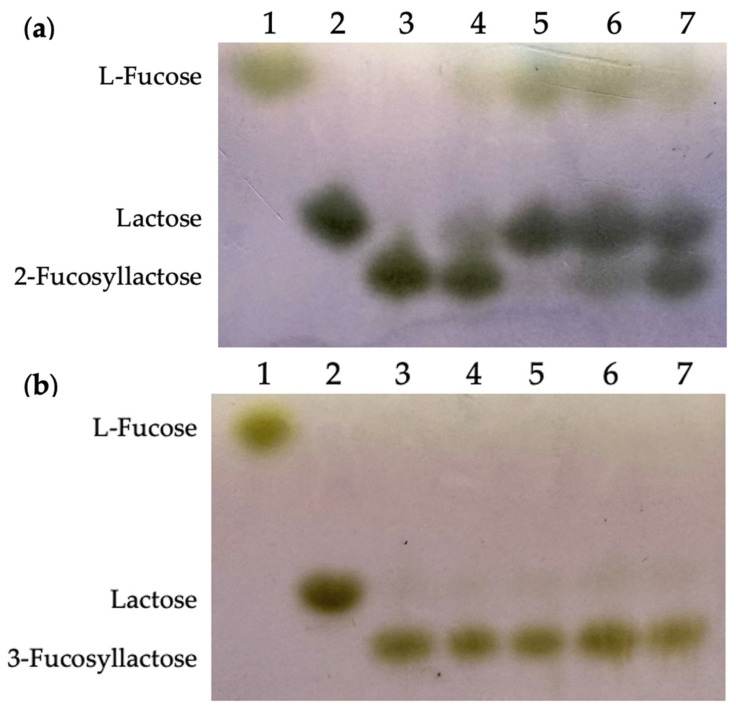
TLC analysis of the 2′-fucosyllactose (**a**) and 3-fucosyllactose (**b**) hydrolysis by α-L-fucosidases. 1—L-fucose, 2—lactose, 3—negative control, 2′-fucosyllactose or 3-fucosyllactose hydrolysed by 4—Fuc25A, 5—Fuc25D, 6—Fuc25E, 7—Fuc25C.

**Table 1 ijms-25-00809-t001:** Physicochemical parameters of the active site pockets for Fuc25A, Fuc25C, Fuc25D, and Fuc25E were evaluated by DoGSiteScorer.

Enzyme	Surface, Å^2^	Volume, Å^3^	Depth, Å	Hydrophobicity	Enclosure
Fuc25A	373.90	267.33	12.34	0.50	0.15
Fuc25C	583.66	723.19	22.28	0.38	0.41
Fuc25D	1128.84	848.64	28.95	0.42	0.12
Fuc25E	647.08	522.03	17.02	0.46	0.17

**Table 2 ijms-25-00809-t002:** Kinetic constants for Fuc25A, Fuc25C, Fuc25D, and Fuc25E. *p*NP-αFuc was used as a substrate.

Enzyme	V_max_, μM s^–1^	K_M_, μM	k_cat_, s^–1^	k_cat_/K_M_, μM^–1^ s^–1^
Fuc25A	1.007 ± 0.040	333.8 ± 21.4	12.08 ± 0.48	0.036 ± 0.003
Fuc25C	0.590 ± 0.096	1401.1 ± 373.6	1.64 ± 0.27	0.001 ± 0.000
Fuc25D	0.504 ± 0.027	79.7 ± 8.6	28.87 ± 1.56	0.364 ± 0.022
Fuc25E	0.305 ± 0.008	85.1 ± 12.9	4.08 ± 0.10	0.049 ± 0.007

**Table 3 ijms-25-00809-t003:** Fucosyl residue acceptors for transfucosylation reaction results observed by HPLC-MS. *p*NP-αFuc was used as a fucosyl donor. The symbol “+” means reaction takes place. The observed molecular ions are provided in Appendix A.

Acceptor Compound	Fuc25A	Fuc25D	Fuc25C	Fuc25E
Monosaccharides				
D-Glucose	+	+	+	+
D-Galactose	+	+	+	+
D-Fructose	+	+	+	+
L-Fucose	Traces *	Traces *	Traces *	+
N-Acetylglucosamine	+	+	+	+
L-Rhamnose	+	+	Traces *	+
D-Ribose	+	+	+	+
D-Xylose	+	+	+	+
D-Mannose	+	+	+	+
Disaccharides				
Lactose	+	+	+	+
Maltose	+	+	+	+
Amino acids				
L-Serine	+	+	+	+
L-Threonine	+	+	+	+
D-Serine	+	+	+	+
D-Threonine	+	+	+	+

* Evaluation “Traces” are assigned to the result table when the corresponding peak area in the reaction mixture sample divided by the corresponding peak area in the negative control sample is >1 but <3.

## Data Availability

All data included in this study are available from the corresponding author by request.

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
