# Peer review of "α-L-Fucosidases from an Alpaca Faeces Metagenome: Characterisation of Hydrolytic and Transfucosylation Potential"

_ijms, 2024, doi:10.3390/ijms25020809_

Round 1

Reviewer 1 Report

Comments and Suggestions for Authors

Summary:

Four new a-L-fucosidases have been identified in a screen of an alpaca feces metagenomic library. The fucosidase genes were inserted into a plasmid and expressed in E coli. Enzymatic parameters of each were characterized using p-nitrophenyl alpha-L-fucopyranoside as substrate. The potential for using the enzymes as transglycosidases was then explored with mono/di/tri/oligosaccharides and some amino acids.

The work is interesting and the manuscript is relatively well organized. There are items that the authors should consider adding into the manuscript and supplementary material, however, before it is accepted for publication.

1. More information on the new fucosidases should be in the text and/or supplementary material.

a. What are the full sequences of the proteins? How will that information be made available on the internet?

b. Can the enzymes be assigned to a species based on the knowledge of the sequence and the known organisms of the alpaca gut microbiome?

c. What is the mechanism of GH29a enzymes? This is an important feature when considering them as potential transglycosidases.

2. The transglycosidation reaction.

Overall the description of the experiments here is incomplete.

a. The experimental for TLC assay are not really discussed in the main text. They should be.

b. The LC-MS experimental is also incomplete. What does the data in the supplemental show? Is it five independent runs? What is the ionization technique? The detection?

c. Transglycosidation with oligosaccharides is presented in the text but not the table. Why? All of the substrates should be in Table 3, with more information (i.e., TLC vs LC-MS, clearer qualitative quantitation, etc.)

3. Minor items

a. References to pNP-Fuc should be consistent to make it clear that the alpha anomer was used.

b. There is only one or two instances in the discussion where the physicochemical properties are considered in light of/along with the observed results. This should be amplified.

Comments on the Quality of English Language

Minor problems that can be corrected by editing.

Reviewer 2 Report

Comments and Suggestions for Authors

This manuscript aims at four α-L-fucosidases screened in the metagenomic library and attempts to characterize the hydrolysis activity of these enzyme samples in α-L-fucosidase, the ability of transfucosylation reactions and activity in the synthesis of fucosylated amino acids. Silico analysis was performed on the manuscript to simulate the structure of the candidate enzyme and further detect the component analysis of its enzyme-catalyzed products.

The experimental data are excellent, and the results and discussions are precise. Suitable for publication with minor modifications.

suggestion

1.      The author can add an explanation as to why he chose the alpaca faeces metagenome.

2.      The supplementary information cited in the text is inaccurate and should be corrected. Line 124, Table S2. Line 349, Table S3 may be appended.

3.      Line 625, "4. Conclusions" is too simple. In addition to responding to the research question or hypothesis, authors should honestly discuss the limitations of the study. At the same time, possible directions for future improvements are proposed, as well as evaluation of contributions to existing research fields or suggestions for future research efforts.
